# Natural Killer Cell Function Tests by Flowcytometry-Based Cytotoxicity and IFN-γ Production for the Diagnosis of Adult Hemophagocytic Lymphohistiocytosis

**DOI:** 10.3390/ijms20215413

**Published:** 2019-10-30

**Authors:** Hyeyoung Lee, Hoon Seok Kim, Jong-Mi Lee, Ki Hyun Park, Ae-Ran Choi, Jae-Ho Yoon, Hyejin Ryu, Eun-Jee Oh

**Affiliations:** 1Department of Laboratory Medicine, Catholic Kwandong University International St. Mary’s Hospital, Incheon 22711, Korea; shomermaid@catholic.ac.kr; 2Department of Laboratory Medicine, Seoul St. Mary’s Hospital, College of Medicine, The Catholic University of Korea, Seoul 06591, Korea; hskim11@catholic.ac.kr (H.S.K.); jongmi1226@naver.com (J.-M.L.); bibichoi69@naver.com (A.-R.C.); 3Department of Biomedicine & Health Sciences, Graduate School, The Catholic University of Korea, Seoul 06591, Korea; neojkl1004@naver.com; 4Department of Hematology, Catholic Hematology Hospital and Leukemia Research Institute, Seoul St. Mary′s Hematology Hospital, College of Medicine, The Catholic University of Korea, Seoul 06591, Korea; royoon@catholic.ac.kr; 5Samkwang Medical Laboratories, Seoul 06742, Korea; hye-jin@catholic.ac.kr

**Keywords:** hemophagocytic lymphohistiocytosis, natural killer cell activity, interferon-gamma, cytotoxicity, cytokine, NK cell subset

## Abstract

Although natural killer (NK) cell function is a hallmark of hemophagocytic lymphohistiocytosis (HLH), there is no standard method or data on its diagnostic value in adults. Thus, we performed a single-center retrospective study of 119 adult patients with suspected HLH. NK cell function was determined using both flowcytometry-based NK-cytotoxicity test (NK-cytotoxicity) and NK cell activity test for interferon-gamma (NKA-IFNγ). NK cell phenotype and serum cytokine levels were also tested. Fifty (42.0%) HLH patients showed significantly reduced NK cell function compared to 69 non-HLH patients by both NK-cytotoxicity and NKA-IFNγ (*p* < 0.001 and *p* = 0.020, respectively). Agreement between NK-cytotoxicity and NKA-IFNγ was 88.0% in HLH patients and 58.0% in non-HLH patients. NK-cytotoxicity and NKA-IFNγ assays predicted HLH with sensitivities of 96.0% and 92.0%, respectively. The combination of NKA-IFNγ and ferritin (>10,000 µg/L) was helpful for ruling out HLH, with a specificity of 94.2%. Decreased NK-cytotoxicity was associated with increased soluble IL-2 receptor levels and decreased CD56dim NK cells. Decreased NKA-IFNγ was associated with decreased serum cytokine levels. We suggest that both NK-cytotoxicity and NKA-IFNγ could be used for diagnosis of HLH. Further studies are needed to validate the diagnostic and prognostic value of NK cell function tests.

## 1. Introduction

Hemophagocytic lymphohistiocytosis (HLH) is a rare disease and a life-threatening syndrome caused by pathological hyperactivation of the immune system. It can lead to hypercytokinemia and multi-organ failure [1]. Adult HLH without genetic predisposition or family history is a rare clinical condition. Its underlying pathophysiological mechanisms have not been fully elucidated yet. HLH is difficult to diagnose because its clinical symptoms and signs are nonspecific, overlapping with those of other diseases such as sepsis, autoimmune disease, multi-organ failure, and progressed malignancies [2]. Not all HLH patients exhibit hemophagocytosis. In addition, this can be observed in other diseases such as severe sepsis, post-transfusion or cytotoxic therapies, and hematologic malignancy [3]. HLH-2004 guideline was developed for the diagnosis of pediatric HLH and commonly used to date for adult HLH diagnosis as well [4,5]. This guideline includes six clinical and laboratory parameters (fever, splenomegaly, cytopenia, hyperferritinemia, hypertriglyceridemia/ hypofibrinogenemia, and hemophagocytosis in bone marrow) and three specialized tests (increased serum soluble interleukin-2 receptor (sIL-2r), decreased natural killer (NK) cell activity, and related gene test), of which 5 or more must be met for a diagnosis of HLH [4,5].

The pathophysiological mechanism of HLH is mainly due to defective function of NK cells and cytotoxic T lymphocytes (CTLs). It results in uncontrolled activation of lymphocytes and macrophages which induce excessive production of cytokines [6]. Therefore, NK cell function test is a hallmark and a valuable tool for HLH diagnosis. Human NK cells can be divided into various subsets by expression of CD16 and CD56 [7]. CD16+CD56dim cells account for majority of NK cells and display cytotoxic activity while CD16-CD56bright cells secrete various cytokines, mainly interferon-gamma (IFNγ) [8]. CD16-CD56bright cells also acquire potent cytotoxicity after activation. NK cells have direct cytotoxicity through granzyme and perforin release. NK cells can also control immune response by cytokine secretion. Therefore, measuring both NK-cytotoxicity and NK cell function for IFNγ release (NKA-IFNγ) might be valuable for the diagnosis of HLH. However, the diagnostic value of NK-cytotoxicity and NKA-IFNγ in HLH patients has not been properly evaluated yet.

To measure NK-cytotoxicity, ^51^Cr-release assay has been known as the gold standard. However, it has many technical shortages such as spontaneous breakdown of ^51^Cr from target cells that could affect analytical sensitivity and radioactive waste generation [9]. Many publications have reported alternative flowcytometric methods. However, flowcytometric-based NK cytotoxic assays have not been standardized across laboratories. They are generally only available in specialized reference centers [10,11,12,13,14,15,16].

NK Vue (ATGen, Seongnam-si, Korea) test is a newly developed in vitro diagnostic assay that can measure NKA-IFNγ using sandwich enzyme-linked immunosorbent assay (ELISA). The principle of this assay is stimulating whole blood with Promoca (engineered recombinant cytokines) that can specifically activate NK cells in whole blood. Released IFNγ level from activated NK cells is then measured [17,18]. Previous studies reported that patients with malignancy show decreased NKA-IFNγ level compared to healthy controls [17,19,20]. However, a study applying this assay in HLH diagnosis has not been reported yet. Thus, the objective of this study was to evaluate flowcytometry-based NK-cytotoxicity and NKA-IFNγ as diagnostic tools for adult HLH. Associations between NK cell function test and clinicopathological parameters including NK cell phenotype and cytokines were also investigated.

## 2. Results

### 2.1. Patients

Of a total of 119 referred adult patients with unexplained fever and cytopenia, 50 (42.0%) patients were clinically diagnosed with HLH based on the HLH-2004 diagnostic criteria. Patients who did not fulfill the HLH-2004 diagnostic criteria were classified as non-HLH group. Clinical and diagnostic laboratory findings are presented in Table 1. The etiologic causes of HLH were leukemia/lymphoma (*n* = 15), infection (*n* = 8, 6 Epstein–Barr virus (EBV), 1 malaria, and 1 fungal infection), autoimmune disease (*n* = 5, 4 systemic lupus erythematosus (SLE), 1 Behcet’s disease), and myelodysplastic syndrome (*n* = 2). We could not identify etiologic causes of HLH in 20 (40.0%) patients. In non-HLH patients, 34 (49.3%) patients had cancer, including hematologic malignancies (*n* = 32) and solid cancers (*n* = 2). The average number of HLH criteria met in the HLH group was 5.6 vs. 3.2 in the non-HLH group (*p* < 0.001). HLH patients presented fever, splenomegaly, hyperferritinemia, hypertriglyceridemia/hypofibrinogenemia, hemophagocytosis, decreased NK-cytotoxicity, and elevated sIL-2r (>2400 U/mL) more frequently than non-HLH patients.

### 2.2. NK Cell Function, NK Cell Subset, and Cytokine Levels in HLH and Non-HLH Patients

All 119 patients with suspected HLH had decreased NK cell function compared to healthy controls. When NK cell functions were compared between HLH and non-HLH patients, HLH patients showed significantly decreased NK-cytotoxicity than non-HLH patients (median (95% confidence interval (CI)): 12.1% (9.6–17.1) vs. 24.3% (16.8–36.3), *p* < 0.001) (Figure 1a). NKA-IFNγ was also significantly decreased in HLH patients than that in non-HLH patients (10.0 pg/mL (10.0–22.8) vs. 34.3 pg/mL (11.7–57.7), *p* = 0.020) (Figure 1b). The number of total NK cell was significantly decreased in HLH patients compared to that in non-HLH patient (Table 2). In addition, CD56dim NK cells (%) decreased in HLH patients compared to those in non-HLH patients (80.9% (65.6-89.7) vs. 91.3% (86.8–94.3), *p* = 0.029). However, percentages of CD56bright, NKG2A (+), or NKG2D (+) NK cells were not significantly different between HLH and non-HLH patients (Appendix A). In cytokine analysis, only serum sIL-2r level was significantly higher in HLH patients than that in non-HLH patients (4433.0 (2482.7–7500.0) vs. 1098.0 (795.9–1653.2), *p* < 0.001) (Appendix A). We normalized NK-cytotoxicity and NKA-IFNγ based on the percentage of NK cells, CD56bright NK cells and CD56dim NK cells (Appendix A). After normalization to CD56dim NK cells, NK-cytotoxicity was observed to be significantly decreased in HLH patients compared with non-HLH patients (21.0 (13.3–24.6) vs. 31.6 (27.0–38.9), *p* < 0.001). However, no significant differences were observed in NKA-IFNγ results.

### 2.3. Comparison of NK-Cytotoxicity and NKA-IFNγ Results

There was no correlation between overall quantitative results from NK-cytotoxicity and NKA-IFNγ (*r* = 0.088, *p* > 0.05). When we evaluate the qualitative results with the cutoff value of 38.5% for NK-cytotoxicity and 250 pg/mL for NKA-IFNγ, agreement between the two assays was 88.0% in HLH group and 58.0% in non-HLH group (*p* < 0.001) (Table 3). Of 29 non-HLA patients showing discrepant results between NK-cytotoxicity and NKA-IFNγ, 21 (72.4%) patients had normal NK-cytotoxicity but decreased NKA-IFNγ levels. In addition, eighteen (62.1%) patients had malignancy and 25 (86.2%) patients had ferritin levels less than 5000 µg/L.

### 2.4. Diagnostic Performance of NK-Cytotoxicity and NKA-IFNγ for HLH

Diagnostic ability of NK-cytotoxicity and NKA-IFNγ for HLH was evaluated in 119 patients with suspicion of HLH (Table 4). NK-cytotoxicity predicted HLH with sensitivity of 96.0% (95% CI; 87.1–99.3%) and specificity of 36.2% (29.8–38.6%) using cutoff value of 38.5% (area under the curve (AUC): 0.696). NKA-IFNγ predicted HLH with sensitivity of 92.0% (83.7–97.3%) and specificity of 17.4% (11.4–21.2%) using cutoff value of 250.0 pg/mL (AUC: 0.619). When we combined these variables, the combination of NKA-IFNγ with ferritin (>10,000 µg/L) improved the specificity up to 94.2%.

### 2.5. Serum Cytokine and NK Cell Subset Results in Association with NK Cell Function

We grouped studied patients according to NK-cytotoxicity and NKA-IFNγ results: decreased NK-cytotoxicity (<38.5%) vs. normal NK-cytotoxicity (>38.5%); decreased NKA-IFNγ (<250 pg/mL) vs. normal NKA-IFNγ (>250 pg/mL). NK cell subset and serum cytokine levels were compared according to NK-cytotoxicity and NKA-IFNγ results (Table 5). Decreased NK-cytotoxicity was associated with decreased total NK cell number, decreased CD56dim NK cells, and increased sIL-2r levels (*p* = 0.003, *p* = 0.015, and *p* = 0.014, respectively). However, the expression of NKG2A or NKG2D was not associated with NK-cytotoxicity results. In terms of NKA-IFNγ, NK cell subset results were not significantly different between decreased- and increased NKA-IFNγ groups. On the other hand, serum cytokine levels were significantly (*p* < 0.05) decreased in patients with decreased NKA-IFNγ (<250 pg/mL) compared to those in patients with normal NKA-IFNγ.

## 3. Discussion

HLH is a life-threatening inflammatory condition. Early diagnosis and intervention are essential for patient survival [1,21]. Among clinical and laboratory parameters in HLH-2004 criteria, three specialized tests (genetic test, serum sIL-2r test, and NK cell function test) are not easy to perform in clinical laboratory. Genetic tests are useful for detecting potential genetic predisposition to HLH in adult patients. However, genetic abnormalities are rarely detected. Pending results may delay clinical decision to treat HLH [22]. Because the hallmark of HLH is deficiency of NK cell function, NK cell function test can be the most valuable parameter for HLH diagnosis [1,23]. This is the first study to assess the diagnostic performance of NK cell function test by both NK-cytotoxicity and NKA-IFNγ in adult HLH patients. We confirmed that both flow cytometry-based NK-cytotoxicity and NKA-IFNγ levels were significantly decreased in HLH patients than those in non-HLH patients.

When we compared NK-cytotoxicity and NKA-IFNγ results, NK-cytotoxicity was not correlated with NKA-IFNγ level. Although the agreement between qualitative results from NK-cytotoxicity and NKA-IFNγ were high in HLH patients (88%), non-HLH patients showed lower agreement (58.0%) between the two NK cell function tests. It might be due to decreased NK cell function in malignancy and increased cytokine levels in infection. Several previous studies have also demonstrated decreased NK cell function in patients with malignancies (including colorectal cancer, prostate cancer, ovarian cancer and hematologic malignancies) compared to healthy controls [17,18,19,24,25,26,27,28,29].

Current HLH diagnosis criteria specify ferritin level > 500 μg/L. However, a recent study reported that the specificity for predicting HLH was as low as 33% with this cut off ferritin level. They suggested that an optimal cutoff level of ferritin at 5775 μg/L for diagnosis of HLH had a sensitivity of 89.5% and a specificity of 63.2% [5]. With cutoff value of 38.5% for NK-cytotoxicity and 250 pg/mL for NKA-IFNγ, the two tests showed sensitivities of 96.0% and 92.0% for diagnosing HLH, respectively. However, specificities of NK-cytotoxicity and NKA-IFNγ were low (36.2% and 17.4%, respectively). To improve the specificity of NK cell function test, we combined NK cell function results and ferritin levels. Combined NKA-IFNγ and ferritin levels (>10,000 µg/L) improved specificity to 94.2%. In addition, combined NK-cytotoxicity, NKA-IFNγ, and ferritin levels (>10,000 µg/L) showed specificity of 95.7%. In practice, there is a trade-off between sensitivity and specificity. Still, no single laboratory parameter is sufficient to diagnosis HLH and superiority of current diagnostic criteria is still investigated. Thus, further new findings could be incorporated in diagnostic and prognostic marker of adult HLH.

Conventional NK cell function assay requires time-consuming cell culturing process and radioactive reagents or specialized flow cytometric techniques [16]. Although flow cytometry-based NK-cytotoxicity has been widely applied and constantly optimized, different laboratories have their own protocols and reference methods. Variations include different effector to target ratios, selection of fluorochromes to label target cells, NK cell purification process using magnetic beads, and the use of rIL-2 to enhance NK cell response [10,11,12,13,14,15,16]. In addition, flow cytometry assays need fresh samples (drawn within 24 h of testing) to be valid. It is known that IFNγ released from NK cell and cytotoxic T cell can activate macrophage to induce secretion of inflammatory cytokines in HLH [30]. As HLH is referred to as a cytokine storm, previous studies have demonstrated elevated levels of inflammatory cytokines including IFNγ and IFNγ-induced chemokines in HLH [31,32]. Among several cytokines, only sIL-2r is included in HLH criteria. Unlike serum cytokine measurements, NKA-IFNγ assay is a functional assay that measures released IFNγ level from activated NK cells. In contrast to NK-cytotoxicity, NKA-IFNγ is a simple, verified test based on a simple commercially available kit. Supernatants containing released IFNγ can be frozen and stored until running the ELISA. Because adult secondary HLH is defined by pathologic immune activation and cytokine release leading to end organ damage, measurement of released IFNγ level from activated NK cells should be highlighted. Although NKA-IFNγ correlates with serum cytokine levels, evaluation of diagnostic performance of NKA-IFNγ is important for clinical implementation.

NK-cytotoxicity level, but not NKA-IFNγ level, correlated with NK cell percentage and NK cell count. Previous study reported NK-cytotoxicity results using isolated NK cells [33]. However, NK-cytotoxicity using peripheral blood mononuclear cells (PBMC) can also provide natural circumstance, like in vivo environment as compared to the test using isolated NK cells. When encountering the target cells, the CD56dim cells secrete the perforin with the granzyme B and then induce cell cytotoxicity. The activated NK cells can also secret IFN-γ and the innate and adaptive immune system is reinforced by interaction of the activated NK cells and other immune cells [34]. Therefore, using PBMC is more convenient and can provide more informative results than using NK cells in clinical laboratory setting [15]. A previous study has revealed that NKA-IFNγ is not associated with NK cell counts in hematologic malignant patients [26]. Based on this finding, we could conclude that decreased level of NKA-IFNγ reflects decreased ability of NK cell function itself, not numerical change of NK cell count. Gao et al. [30] have found that secondary HLH patients have increased expression of inhibitory NKG2A receptor and decreased expression of activating NKG2D. Changes of NK cell receptor and its interaction with adaptors can impair the cytotoxic capacity of NK cells against pathogenic T cells. Therefore, hyperactive T cells continue to activate macrophage and lead to cytokine storm [30]. In our data, patients with decreased NK-cytotoxicity showed tendency of increased expression of NKG2A but decreased expression of NKG2D than patients with normal NK-cytotoxicity, although these differences were not statistically significant. In addition, the expression of NKG2A or NKG2D was not associated with NKA-IFNγ results. NKA-IFNγ does not measure cytotoxicity capacity, but focuses on the regulatory capacity as IFNγ producers [35]. Therefore, it is unlikely to be associated with NK cell receptors. In previous study, Mahapatra S. et al. [36], measured both the percentage and median fluorescent intensity (MFI) of NK cell phenotypes in healthy adults and children. They reported that significantly different NK markers were not consistent between percentage and MFI and emphasized the importance of using both measures. While percentages measure the frequency of positively expressing NK cells at a population level, MFI reflect the molecules per cell for markers expressed by individual NK cells. Thus, further studies are needed to analyze the MFI of NK cell phenotypes in association with NK function in HLH patients. When we normalize NK-cytotoxicity data to CD56dim NK cells (%), HLH patients showed significantly decreased NK-cytotoxicity levels. However, normalized NKA-IFNγ results did not show significant differences between HLH and non-HLH patients. Further validation studies are needed for NKA-IFNγ tests in association with NK subpopulation.

This study has several limitations. First, this retrospective study was performed in a single institution. We did not simultaneously perform genetic test and sIL-2r tests for all our study population. Recently, serum sIL-2r has been reported as a good test with low cost for adult HLH [5]. However, in the present study, sIL-2r assay only showed sensitivity of 69.6% in HLH patients who were available for the test. It might be due to different population and etiologic causes of HLH. In addition, we did not assess prognostic value of NK cell function test. Although our data originated from a retrospective study, we could assess NK cell functions in all patients with suspected HLH. HLH is a rare disease and hematologic malignancies are reported to be the main trigger of HLH [3]. If clinicians do not have clinical suspicion of HLH and lack specific knowledge about HLH, the prevalence of HLH may be under- or overestimated, especially in adult patients [3,21]. In the present study, all patients enrolled in this study had fever and cytopenia, therefore, the incidence of disease showed higher than expectation. We have evaluated the performance of newly developed NK function assay for diagnosis HLH. Diagnostic values of NKA-IFNγ should be validated by larger studies. In addition, further study is needed to compare NKA-IFNγ with NK-cytotoxicity using isolated NK cells with or without C107a expression. It might be worthwhile to conduct the intracellular cytokine staining to compare the predictability of NK function assay [37,38].

In conclusion, our results suggest that both NK-cytotoxicity and NKA-IFNγ might be useful for HLH diagnosis. We expect that NKA-IFNγ assay is a convenient and supportive screening test for HLH. Compared to NK-cytotoxicity, NKA-IFNγ showed similar performance for predicting HLH. It also showed high level of agreement with NK-cytotoxicity. Further validation studies are needed for NK cell function tests in large population and the prognostic value of NKA-IFNγ.

## 4. Materials and Methods

### 4.1. Patients

A total of 119 adult patients with clinically suspected HLH were referred for NK cell function test in Seoul St. Mary’s Hospital from November 2015 to December 2018. HLH was suspected by physicians in cases of an unexplained fever and cytopenia. The confirmation of a suspected HLH case was performed by the diagnostic criteria in HLH-2004 guideline. All patients presented with fever or cytopenia. We finally diagnosed HLH among those patients who met 5 or more HLH 2004 criteria (1, 4). Parameters included in the diagnostic guidelines for HLH were fever, splenomegaly, cytopenia affecting at least two lineages, triglycerides, fibrinogen, ferritin, and NK-cytotoxicity. Genetic test was not done and sIL-2r test was available in 39 patients. This study was approved by the Institutional Review Board of Seoul St. Mary’s Hospital (KC15TISE0936; approved date: 16 February 2016) and (KC19RESI0632; approved date: 18 September 2019). Of 119 adult patients, informed consent was received from 50 HLH patients. Informed consent from non-HLH patients was waived by the board (KC19RESI0632) because the current retrospective study was performed using medical records. As a control for NK-cytotoxicity and NKA-IFNγ assay, 78 healthy donors without a history of autoimmune disorders, major infection, and other inflammatory diseases were included.

### 4.2. NK-Cytotoxicity Assay

CML-derived K562 cell line was cultured with Dulbecco′s Modified Eagle′s Medium (DMEM, Welgene, Daegu, Korea) supplemented with 100 U/mL penicillin (Gibco, BRL, NY, USA), 100 μg/mL streptomycin (Gibco), and 10% fetal bovine serum. PBMCs were isolated from heparinized blood using Uni-sep lymphocyte separation tubes (Novamed, Jerusalem, Israel) within 4 h after heparinized blood collection. NK-cytotoxicity assay was performed as previously reported [15]. Briefly, K562 cells were stained with carboxyfluorescein succinimidyl ester (CFSE) to label target cells from effector cells. Effector cells (PBMC) were incubated at 37 °C with CFSE labeled K562 cells at effector to target ratios of 32:1 in 96-well plates for 4 h. For negative control, K562 cells alone without effector cells were used in each test. After 4 h of culture at 37 °C with 5% CO_2_, cell mixture was stained with 7-AAD (Beckman Coulter Life Sciences, Indianapolis, IN, USA) and analyzed with a Navios flow cytometer (Beckman Coulter, Miami, FL, USA). NK-cytotoxicity levels were determined by calculating positive cells for 7-AAD in CFSE positive cells. Based on results of the normal controls, cutoff% for decreased NK-cytotoxicity in our center was set to be <38.5%.

### 4.3. NKA-IFNγ Assay

NKA-IFNγ level was measured by enzyme immunoassay using NK Vue-Kit (ATGen, Seongnam-si, Korea) as previously described [17,26]. Fresh whole blood (1 mL) was collected using NK Vue tube containing Promoca, a stimulatory cytokine that could specifically stimulate NK cells. According to the previous report, the major cell population secreting IFNγ after stimulating whole blood with Promoca was NK cells [18]. Nederby L. et al., performed intracellular flow cytometry and proved that the predominant source of IFNγ in this assay was NK cells [35]. Minor faction of secreted IFNγ was from T cell and NKT cells, but they concluded the secreted IFNγ after incubated with Promoca was reflection of NK cell activity in whole blood. Tubes were incubated at 37 °C for 20–24 h. Cell-free supernatants were then transferred and stored at −70 °C until IFNγ levels were measured using the ELISA method according to the manufacturer’s instructions. Briefly, 50 μL of samples, controls, and six standards were incubated at room temperature for 2 h on anti-human IFNγ coated plates and washed with wash buffer. IFNγ conjugate was added and incubated at room temperature for 1 h. After washing, plates were incubated with 100 μL of substrate in the dark for 30 min at room temperature. Optical density was then measured at a wavelength of 450 nm. The concentration of IFNγ was determined using a calibration curve. The measurement range was 10–4000 pg/mL and the limit of detection of this assay was 10 pg/mL. Total inaccuracy of two levels of control had coefficient of variation < 15%.

### 4.4. NK cell Subset Analysis

Whole blood (100 μL) was stained with monoclonal antibodies. Anti-CD45-FITC (clone; HI30), anti-CD3-V450 (clone; UCHT1), anti-NKGD-APC (clone; 1D11), and anti-CD56-PE-Cy7 were purchased from Becton Dickinson (San Jose, CA, USA). Anti-NKG2A–PE was obtained from Miltenyi Biotec (Bergisch Gladbach, Germany). Data were acquired using a Fortessa flowcytometry (BD Biosciences, San Jose, CA, USA) and analyzed using FlowJo version 10.0.6 software (Tree Star, Ashland, OR, USA) (Figure 2). Lymphocyte count in whole blood was measured with Sysmex XN (Sysmex, Kobe, Japan).

### 4.5. Cytokine

Concentrations of cytokines (IL-2, IL-6, IFNγ, TNFα) and chemokine (CXCL10) were measured with MILLIPLEX MAP human cytokine/chemokines panel (Millipore, Schwalbach/Ts, Germany) using Luminex 200 instrument (Luminex, Austin, TX, USA) according to manufacturers’ instructions. Serum sIL-2r levels were measured with an enzyme-linked immunosorbent assay (Siemens IMMULITE Immunoassay platform; adult reference range, 241–846 U/mL).

### 4.6. Statistical Analysis

Statistical analyses were performed using MedCalc Statistical Software version 19.0.3 (MedCalc Software bvba, Ostend, Belgium). Results are presented as mean ± standard deviation (SD) or median with 95% CI. The Mann–Whitney test was used for comparison of median values and Chi-square was used to evaluate differences of proportions. A *p* value of less than 0.05 was considered statistically significant.

## Figures and Tables

**Figure 1 ijms-20-05413-f001:**
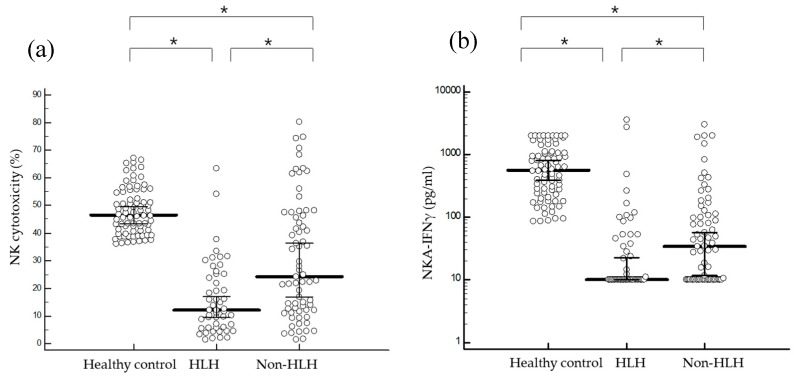
Comparison of NK-cytotoxicity and NKA-IFNγ results. (**a**) NK-cytotoxicity levels were significantly decreased in HLH patients (median (95% CI): 12.1% (9.6–17.1)) compared to those in non-HLH patients (24.3% (16.8–36.3)) (*p* < 0.001) or healthy controls (46.6% (43.5-49.7)) (*p* < 0.001). (**b**) NKA-IFNγ values were significantly decreased in HLH patients (10.0 pg/mL (10.0–22.8)) than those in non-HLH patients (34.3 pg/mL (11.7–57.7)) (*p* = 0.020) and healthy controls (564.5 pg/mL (391.1–814.5)) (*p* < 0.001). * *p* < 0.05.

**Figure 2 ijms-20-05413-f002:**
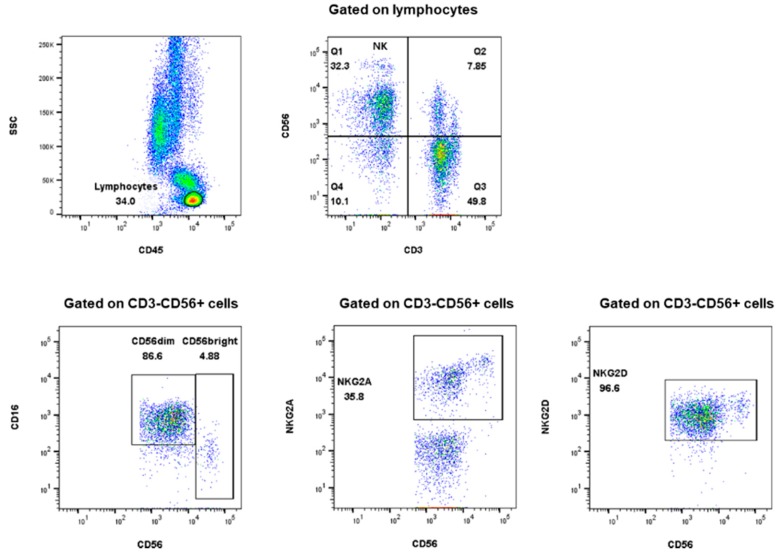
Gating strategy for NK cell subset analysis.

**Table 1 ijms-20-05413-t001:** Clinical characteristics of the study population

	HLH (*n* = 50)	Non-HLH (*n* = 69)	*p* Value
Men, *n* (%)	28 (56.0)	36 (52.2)	NS
Age at diagnosis, year (mean +/− SD)	51.6 +/− 16.4	47.1 +/− 19.0	NS
Etiological causes, *n* (%)			
Leukemia/Lymphoma	15 (30.0)	21 (30.4)	NS
Other hematologic malignancy	2 (4.0)	11 (15.9)	0.040
Solid cancer	0 (0.0)	2 (2.9)	NS
Infection	8 (16.0)	13 (18.8)	NS
Autoimmune disease	5 (10.0)	14 (20.3)	NS
Unknown	20 (40.0)	8 (11.6)	<0.001
HLH diagnostic criteria, *n* (%)			
Fever (>38.5 °C), *n* (%)	49 (98.0)	51 (73.9)	<0.001
Splenomegaly, *n* (%)	31 (62.0)	13 (18.8)	<0.001
Hemoglobin, g/L (median, 95% CI)	9.7 (9.1–10.2)	9.9 (9.3–10.9)	NS
Platelet, 100 × 10^9^/L (median, 95% CI)	75 (55.2–10.6)	77 (59.0–107.1)	NS
Absolute neutrophil count, 1.0 × 10^9^/L (median, 95% CI)	1855.0 (1106.0–2855.2)	1670.0 (1175.5–2971.1)	NS
Triglycerides, mg/dl (median, 95% CI)	203.5 (160.0–237.2)	153.0 (131.0–175.9)	0.016
Fibrinogen, g/L (median, 95% CI)	256.0 (171.0–333.9)	319.5 (258.8–351.9)	NS
Ferritin, μg/L (median, 95% CI)	5452.5 (2589.6–9752.4)	1786.0 (1182.1–2608.6)	<0.001
Hemophagocytosis in bone marrow, *n* (%)	43 (86.0)	14 (20.3)	<0.001
Soluble IL-2 receptor, U/mL (*n* = 39) (median, 95% CI)	4433.0 (2482.7–7500.0)	1098.0 (795.9–1653.2)	<0.001
NK-cytotoxicity, % (median, 95% CI)	12.1 (9.6–17.1)	24.3 (16.8–36.3)	<0.001
No. of HLH criteria fulfilled (mean+/−SD)	5.6 +/− 1.0	3.2 +/− 1.1	<0.001
Overall mortality, *n* (%)	12 (24.0)	12 (17.4)	NS

NS: not significant; CI: confidence interval; *n*, number

**Table 2 ijms-20-05413-t002:** NK cell subset and cytokine levels in patients with HLH and non-HLH patients.

	HLH (*n* = 50)	Non-HLH (*n* = 69)	*p* Value
WBC, N per mL	4040.0 (2507.3–6331.3)	3660.0 (2522.6–5154.9)	NS
Lymphocyte, %	18.0 (13.0–24.8)	27.3 (21.9–31.0)	NS
Lymphocyte, N per mL	565.0 (459.4–917.3)	891.5 (694.9–1099.3)	NS
NK cell, %	6.8 (4.1–8.9)	9.8 (7.7–13.6)	0.026
NK cell, N per mL	41.5 (19.2–64.9)	105.0 (54.4–141.5)	0.014
NK cell subset analysis			
CD56bright, %	1.1 (0.5–4.5)	2.0 (1.1–3.3)	NS
CD56dim, %	80.9 (65.6–89.7)	91.3 (86.8–94.3)	0.029
CD56(−)CD16(+), %	6.9 (2.1–9.9)	10.5 (6.5–13.1)	NS
NKG2A, %	46.5 (30.6–54.1)	41.5 (27.9–53.8)	NS
NKG2D, %	78.9 (67.3–88.6)	86.2 (73.8–91.9)	NS
Cytokine			
CXCL10, pg/mL	1677.0 (1131.7–3999.2)	2361.5 (424.5–5848.4)	NS
IFN-r, pg/mL	13.2 (3.6–39.5)	8.0 (2.6–26.1)	NS
IL-2, pg/mL	1.5 (0.4–2.4)	0.9 (0.0–1.6)	NS
IL-6, pg/mL	4.1 (1.6–20.6)	0.5 (0.0–13.9)	NS
sIL-2r, IU/mL (*n* = 39)	4433.0 (2482.7–7500.0)	1098.0 (795.8–1653.2)	<0.001
TNF-α, pg/mL	21.3 (12.6–54.1)	18.2 (9.6–45.4)	NS

NS: not significant. All values are presented as median (95% confidence interval).

**Table 3 ijms-20-05413-t003:** Agreement between NK-cytotoxicity and NKA-IFNγ results in patients with suspicious HLH.

	NK-Cytotoxicity/ NKA-IFNγ, *n* (%)				Agreement (%)
	Decreased ^a^/Decreased ^b^	Normal/Decreased	Decreased/Normal	Normal/Normal	
HLH (*n* = 50)	44 (88.0)	2 (4.0)	4 (8.0)	0 (0.0)	88.0
Non-HLH (*n* = 69)	36 (52.2)	21 (30.4)	8 (11.6)	4 (5.8)	58.0
All subjective (*n* = 119)	80 (67.2)	23 (19.3)	12 (10.1)	4 (3.4)	70.6

^a^ < 38.5%, ^b^ < 250 pg/mL.

**Table 4 ijms-20-05413-t004:** Diagnostic accuracy of NK-cytotoxicity, NKA-IFNγ, and ferritin for HLH diagnosis.

In Patients Suspicious of HLH (*n* = 119)	Sensitivity (95% CI)	Specificity (95% CI)
NK-cytotoxicity < 38.5%	96.0 (87.1–99.3)	36.2 (29.8–38.6)
NKA-IFNγ < 250 pg/mL	92.0 (83.7–97.3)	17.4 (11.4–21.2)
ferritin > 500 µg/L	98.0 (90.4–99.9)	18.8 (13.4–20.2)
ferritin > 10,000 µg/L	36.0 (26.3–43.3)	89.9 (82.9–95.1)
NK-cytotoxicity < 38.5% + ferritin > 10,000 µg/L	34.0 (25.0–39.3)	94.2 (87.7–98.0)
NKA-IFNγ < 250 pg/mL + ferritin > 10,000 µg/L	30.0 (21.2–35.3)	94.2 (87.9–98.0)
NK-cytotoxicity < 38.5% + NKA-IFNγ < 250 pg/mL + ferritin > 10,000 µg/L	28.0 (19.6–32.4)	95.7 (89.6–98.8)

**Table 5 ijms-20-05413-t005:** NK cell subset and serum cytokine levels in association with NK cell function test results.

	NK-Cytotoxicity			NKA-IFNγ		
	>38.5%	<38.5%	*p*-Value	>250 pg/mL	<250 pg/mL	*p*-Value
WBC, N per mL	3920 (2433.8–5981.2)	3720 (2615.0–5090.0)	NS	2365 (1445.0–5258.1)	4200 (3190.0–5388.2)	NS
Lymphocyte, %	27.3 (21.9–35.3)	21.8 (18–26)	NS	41.0 (21.2–63.5)	22.0 (18–25.7)	0.041
Lymphocyte, N per mL	791 (714.7–1357.3)	721 (525.1–930.0)	NS	840.5 (440.1–1179.4)	769 (557.4–933.6)	NS
NK cell, %	13.9 (9.8–19.0)	6.7 (5.4–8.5)	0.001	9.2 (3.2–14.7)	8.2 (6.7–9.9)	NS
NK cell, N per mL	130 (100.7–152.5)	41 (28.6–72.2)	0.003	50.5 (16.8–147.2)	66 (39.3–103.1)	NS
**NK subset analysis**						
CD56bright, %	0.9 (0.3–2.0)	2.5 (1.1–4.5)	0.034	2.1 (0.4–8.7)	1.6 (0.6–3.7)	NS
CD56dim, %	93.8 (90.1–95.9)	84.1 (73.9–89.8)	0.015	89.7 (76.7–94.1)	87.7 (75.2–92.2)	NS
CD56(−)16(+), %	11.3 (5.7–14.5)	7.1 (3.2–9.9)	NS	4.5 (0.7–11.9)	8.8 (6.4–11.4)	NS
NKG2A, %	39.2 (26.8–56.6)	43.8 (32.3–54.0)	NS	43.5 (14.8–64.5)	43.8 (32.2–53.9)	NS
NKG2D, %	85.9 (70.3–91.3)	81.0 (69.3–89.4)	NS	92.3 (67.3–95.1)	81.0 (68.9–88.2)	NS
**Cytokine**						
CXCL, pg/mL	1235.5 (155.7–11504.7)	1886.0 (1133.5–4491.3)	NS	9844.5 (4703.1–13800.9)	155.0 (729.2–2308.4)	0.001
IFN-r, pg/mL	7.7 (0.5–91.3)	12.6 (3.6–22.7)	NS	159.3 (37.5–1524.5)	8.2 (3.1–12.9)	<0.001
IL-2, pg/mL	0.0 (0.0–0.9)	1.5 (0.9–2.0)	0.015	1.8 (1.2–62.6)	1.0 (0.0–1.4)	0.028
IL-6, pg/mL	0.3 (0.0–61.9)	3.9 (0.9–15.3)	NS	19.4 (2.4–613.5)	1.3 (0.5–5.6)	0.037
sIL-2r, IU/mL	906.0 (528.2–2681.3)	2882.0 (1654.7–6281.9)	0.014	6226.0 (1759.8–16460.9)	1785.0 (1123.1–3112.0)	0.037
TNF-α, pg/mL	14.7 (2.3–37.0)	22.2 (12.7–51.0)	NS	47.1 (17.5–165.8)	18.1 (10.9–30.4)	NS

NS: not significant. All values are presented as median (95% confidence interval).

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
