# Peer review of "Natural Killer Cell Function Tests by Flowcytometry-Based Cytotoxicity and IFN-γ Production for the Diagnosis of Adult Hemophagocytic Lymphohistiocytosis"

_ijms, 2019, doi:10.3390/ijms20215413_

Round 1

Reviewer 1 Report

The authors aim to develop an NK cell based assay that can diagnose the rare disease, Hemophagocytic lymphohistiocytosis (HLH) from other diseases that have similar and overlapping clinical symptoms. This study is clearly within the aims and scope of the journal and such an assay would have clear clinical benefit for the diagnosis of HLH as well as non-HLA patients.

Feedback:

Page 2, line 82 – is there a better way of describing the clinical diagnosis of HLH and in table 1? e.g. instead of ‘causes of HLH’ perhaps ‘etiological causes of HLH’ is more appropriate since e.g. EBV alone doesn’t cause HLH?

Patients from the different groups will differ in the % of CD56-dim ‘cytotoxic’ NK cells in the PBMCs used in the cytotoxicity assay which will profoundly affect the results in this assay i.e. patients with higher % of CD56-dim NK cell in their PBMC will be expected to have higher cytotoxicity than patients with a lower % of CD56-dim NK cells in PBMC in the cytotoxicity assays. In this regard, the authors found that patients with decreased numbers of NK cells had decreased NK cell cytotoxicity (page 6, line 142-143). Can the authors control for this by normalising for the amount of CD56-dim NK cells in each patient’s PBMC since they have access to this flow cytometry data? Perhaps if they perform this analysis they might see a greater difference between NLH and non-HLH groups?

This will also be case for the cytokine secretion assay with the caveat that both CD56-dim and CD56-bright NK cells can produce cytokines it is the CD56 bright NK that produce the most - so can the authors normalise for total NK, CD56-dim only, and CD56-bright only for the IFN-gamma secretion assays? If so, they might see a greater difference between NLH and non-HLH groups for IFN-gamma secretion?

What is ‘PROMOCA’. It is described as “engineered recombinant cytokines” (I suspect IL-12 and IL-18) but in order to properly review this work we need to know what those cytokines are and how they are specific for NK cells and not e.g. T cells that also produce IFN-g and TNF-a.

A much easier assay might be to look at NK cell degranulation using CD107a staining as a general marker of NK cell activation – did the authors perform this staining? It would seem to be much simpler than the assay performed above?

Author Response

Reviewer 1.

The authors aim to develop an NK cell based assay that can diagnose the rare disease, Hemophagocytic lymphohistiocytosis (HLH) from other diseases that have similar and overlapping clinical symptoms. This study is clearly within the aims and scope of the journal and such an assay would have clear clinical benefit for the diagnosis of HLH as well as non-HLA patients.

Feedback:

Page 2, line 82 – is there a better way of describing the clinical diagnosis of HLH and in table 1? e.g. instead of ‘causes of HLH’ perhaps ‘etiological causes of HLH’ is more appropriate since e.g. EBV alone doesn’t cause HLH?

According to the reviewer`s comment, we changed “causes” into “etiologic causes” in overall manuscript and table.

Patients from the different groups will differ in the % of CD56-dim ‘cytotoxic’ NK cells in the PBMCs used in the cytotoxicity assay which will profoundly affect the results in this assay i.e. patients with higher % of CD56-dim NK cell in their PBMC will be expected to have higher cytotoxicity than patients with a lower % of CD56-dim NK cells in PBMC in the cytotoxicity assays. In this regard, the authors found that patients with decreased numbers of NK cells had decreased NK cell cytotoxicity (page 6, line 142-143). Can the authors control for this by normalising for the amount of CD56-dim NK cells in each patient’s PBMC since they have access to this flow cytometry data? Perhaps if they perform this analysis they might see a greater difference between NLH and non-HLH groups?

This will also be case for the cytokine secretion assay with the caveat that both CD56-dim and CD56-bright NK cells can produce cytokines it is the CD56 bright NK that produce the most - so can the authors normalise for total NK, CD56-dim only, and CD56-bright only for the IFN-gamma secretion assays? If so, they might see a greater difference between NLH and non-HLH groups for IFN-gamma secretion?

Answer and correction;

Thank you for careful review. However, NK cytotoxicity assay is highly non-linear and proportional response, therefore normalizing the results for the amount of CD56-dim NK cells may be not appropriate and the conclusions can be misleading. According to the reviewer’s comments, we added discussion as “Previous study reported NK cytotoxicity results using isolated NK cells (33). However, NK cytotoxicity using PBMC can also provide natural circumstance, like in vivo environment as compared to the test using isolated NK cells. When encountering the target cells, the CD56dim cells secrete the perforin with the granzyme B and then induce cell cytotoxicity. The activated NK cells can also secret IFN-γ and the innate and adaptive immune system is reinforced by interaction of the activated NK cells and other immune cells (34). Therefore, utilizing PBMC is more convenient and can provide more informative results than utilizing NK cells in clinical laboratory setting (15).” in lines 217-223.

What is ‘PROMOCA’. It is described as “engineered recombinant cytokines” (I suspect IL-12 and IL-18) but in order to properly review this work we need to know what those cytokines are and how they are specific for NK cells and not e.g. T cells that also produce IFN-g and TNF-a.

Answer and correction;

Promoca is proprietary NK stimulating cytokine cocktail. However, the exact composition of Promoca is not informed by the manufacturer. According to the reviewer`s comment, we added more explanations about the NKA-IFNg assay in line 298-302 as “According to the previous report, the major cell population secreting IFNg after stimulating whole blood with Promoca was NK cells (18). Nederby L. et al., performed intracellular flow cytometry and proved that the predominant source of IFNg in this assay was NK cells (35). Minor faction of secreted IFN was from T cell and NKT cells, but they concluded the secreted IFN after incubated with Promoca was reflection of NK cell activity in whole blood.”

A much easier assay might be to look at NK cell degranulation using CD107a staining as a general marker of NK cell activation – did the authors perform this staining? It would seem to be much simpler than the assay performed above?

Answer and correction;

We agree with reviewer`s opinion. However, in present study, we did not perform the CD107a staining. In previous study [Park KH el al., BioMed research international. 2013;2013:210726.), we compared CD107a expression and NK cytotoxicity and reported that there was a weak correlation between CD107a expression and flowcytometry based NK cell cytotoxicity (r=0.3, P<0.001). Because NK cell cytotoxicity is a stepwise combined process including adhesion, activation and secretion of lytic granules, CD107a expression may not necessarily correlate to NK cytotoxicity.

According to the reviewer`s comment, we added suggestion about the CD107a expression in discussion section in line 256-257 as “In addition, further study is needed to compare NKA-IFNr with NK cytotoxicity using isolated NK cells with or without C107a expression.”

Reviewer 2 Report

Dear authors,

I have read and reviewed your submission "Natural Killer Cell Function Tests by Flowcytometry-based Cytotoxicity and IFN-gamma Production for the Diagnosis of Adult Hemophagocytic Lymphohistiocytosis" and I am sending my comments below.

The study aims to add to the existing diagnostic tools applicable to Adult Hemophagocytic Lymphohistiocytosis" with information on combined used of flow cytometry based Ifng and cytotoxicty in a cohort of chosen patients. As a reviewer, my major concern is the difficulty when judging an article such as this one without any real data figure to show where the numbers in tables come from. It is essential to show data and explaining on cut-off decisions so that readers and reviewers may have an opinion on the capacity to implement such tests in their own labs, even if cut-off values are clearly mentioned in the text. This can eventually be supplied as supplementary information or as a representative plot next to the first tables presented.

If such data are convincing, then the article presents a study conducted with a cohort of patients of sufficient size and interest to support claims and may be of use for the community. However, as the authors state, the study does present important limitations and requires additional validation in other centers.

Unfortunately, the IFNg test seems limited in the capacity to discriminate samples with low levels of IFNg.  It is hard to believe that so many samples measure exactly 10pg/ml. This affects the whole article as an article with this title cannot be based on a test that fails to accurately measure (it is stated that it can measure from 0.1-4000 pg/ml, but data shown raise many doubts) and this is the basis of my decision to reject the article. Nevertheless, I am willing to carefully consider authors explanations and eventually reverse this decision.

More specific comments:

1-Some data are as of now not presented for myself or readers to form an opinion. This is very clear when NK cell phenotype (that include definitions based on intensity) is just not shown at all. 

2- Why are so many samples measured at the same exact values of IFNg (10pg/ml)? Does this reflect limit of detection of test? what values are found for  a "blank" sample? how does this affect the test?

3- It is very difficult to interpret data from the NKA data from figure 2 as the numbers of samples above threshold is too low and numbers in the different conditions significantly differ.  

Minor points:

1- Graph design is not appealing, dots are probably too small. (especially in figure 2). Does it look better with slightly bigger dots?

Best regards,

Author Response

Reviewer 2

I have read and reviewed your submission "Natural Killer Cell Function Tests by Flowcytometry-based Cytotoxicity and IFN-gamma Production for the Diagnosis of Adult Hemophagocytic Lymphohistiocytosis" and I am sending my comments below.

The study aims to add to the existing diagnostic tools applicable to Adult Hemophagocytic Lymphohistiocytosis" with information on combined used of flow cytometry based Ifng and cytotoxicty in a cohort of chosen patients. As a reviewer, my major concern is the difficulty when judging an article such as this one without any real data figure to show where the numbers in tables come from. It is essential to show data and explaining on cut-off decisions so that readers and reviewers may have an opinion on the capacity to implement such tests in their own labs, even if cut-off values are clearly mentioned in the text. This can eventually be supplied as supplementary information or as a representative plot next to the first tables presented.

Answer and correction;

Thank you for careful review. According to the reviewer`s comment, we added supplementary figure 1, 2 and 3.

If such data are convincing, then the article presents a study conducted with a cohort of patients of sufficient size and interest to support claims and may be of use for the community. However, as the authors state, the study does present important limitations and requires additional validation in other centers.

Unfortunately, the IFNg test seems limited in the capacity to discriminate samples with low levels of IFNg. It is hard to believe that so many samples measure exactly 10pg/ml. This affects the whole article as an article with this title cannot be based on a test that fails to accurately measure (it is stated that it can measure from 0.1-4000 pg/ml, but data shown raise many doubts) and this is the basis of my decision to reject the article. Nevertheless, I am willing to carefully consider authors explanations and eventually reverse this decision.

Answer and correction;

Thank you very much for pointing out this error. Although the analytic measurement range of NKA-IFNr assay was 0.1 -4000 pg/mL according to the manufacturer's instructions, the limit of detection of this assay was 10 pg/ml in our center. We made the correction and clarified that in lines 309-310 as “The measurement range was 10-4000 pg / mL and the limit of detection of this assay was 10 pg/mL.”

More specific comments:

1-Some data are as of now not presented for myself or readers to form an opinion. This is very clear when NK cell phenotype (that include definitions based on intensity) is just not shown at all.

Answer and correction;

We agree with reviewer`s opinion. However, in present study, we determined percentage of the cell populations instead of fluorescent intensity. We added the limitation of this study in discussion section (lines of in 236-242) as “In previous study, Mahapatra S. et al.(36), measured both the percentage and median fluorescent intensity (MFI) of NK cell phenotypes in healthy adults and children. They reported that significantly different NK markers were not consistent between percentage and MFI and emphasized the importance of utilizing both measures. While percentages measure the frequency of positively expressing NK cells at a population level, MFI reflect the molecules per cell for markers expressed by individual NK cells. Thus, further studies are needed to analyze the MFI of NK cell phenotypes in association with NK function in HLH patients.”

2- Why are so many samples measured at the same exact values of IFNg (10pg/ml)? Does this reflect limit of detection of test? what values are found for a "blank" sample? how does this affect the test?

Answer and correction;

As we answered above, the limit of detection of this assay was 10 pg/ml in our center. We made the correction and clarified that in lines 309-310 as “The measurement range was 10-4000 pg / mL and the limit of detection of this assay was 10 pg/mL.”

3- It is very difficult to interpret data from the NKA data from figure 2 as the numbers of samples above threshold is too low and numbers in the different conditions significantly differ. 

Answer and correction;

We agree with the reviewer. We deleted the Figure 2 and the results were described in the results section only.

Minor points:

Graph design is not appealing, dots are probably too small. (especially in figure 2). Does it look better with slightly bigger dots?

Answer and correction;

We modified figure 1 with bigger dots.

Reviewer 3 Report

The authors describe the pairing of a flow cytometry-based cytotoxicity test with IFNg production by NK cells to diagnose HLH. While overall the study suggests that these two tests could separate HLH from other samples, it's not clear that they could actually diagnose this disease. This mainly stems from the lack of description of the sample population that is being used. In the introduction the authors list a number of criteria for HLH but not which ones they are using for this study. In addition, for the patient data in Table 1, the distinction between HLH and non-HLH seems to be misleading. In the methods section, it implies that all of these patients were suspected of having HLH, having met "some criteria of HLH 2004." If they are all suspected of having HLH, how can you confirm that this combination in this study is actually diagnosing HLH since 70% of the HLH patient population has either lymphoma or "unknown" etiology. There is also no normal or non-suspect population for comparison although those are presented in the figures but not described in the methods.

There are several contradictory statements throughout the manuscript:
Line 63: Many publications have reported
64 alternative flowcytometric methods. However, flowcytometric-based NK cytotoxic assays have not
65 been standardized across laboratories. They are not available outside of specialized reference centers

If they are unavailable, what value does it add to use them in this manner?

Line 79: Of a total of 119 adult patients, 50 (42.0%) patients were clinically diagnosed with HLH based
80 on the HLH-2004 diagnostic criteria.

The introduction states this as a rare disease, but 42% is not rare.

Line: 115 There was no correlation between quantitative results from NK-cytotoxicity and NKA-IFN (r =
116 0.088, p > 0.05). With cutoff value of 38.5% for NK-cytotoxicity and 250 pg/mL for NKA-IFN,
117 agreement between the two assays

Is there correlation or not?

Lastly, there are several improvements needed to make the manuscript more readable.

1) Table 1: Identify what the values are that are presented, see Table.

2) Table 1: Causes seems to be the wrong wording since you can't have a cause for a non-HLH "disease"

3) Although senestitivity is presented in Table 4, no where in the manuscript is sensitivity discussed. The authors focus solely on the specificity which when using these two methods plus ferritin (where did that come from) increases to 95% specificity but reduces sensitivity to 28%. That trade-off must be discussed and justified.

Author Response

Reviewer 3

The authors describe the pairing of a flow cytometry-based cytotoxicity test with IFNg production by NK cells to diagnose HLH. While overall the study suggests that these two tests could separate HLH from other samples, it's not clear that they could actually diagnose this disease. This mainly stems from the lack of description of the sample population that is being used. In the introduction the authors list a number of criteria for HLH but not which ones they are using for this study. In addition, for the patient data in Table 1, the distinction between HLH and non-HLH seems to be misleading. In the methods section, it implies that all of these patients were suspected of having HLH, having met "some criteria of HLH 2004." If they are all suspected of having HLH, how can you confirm that this combination in this study is actually diagnosing HLH since 70% of the HLH patient population has either lymphoma or "unknown" etiology. There is also no normal or non-suspect population for comparison although those are presented in the figures but not described in the methods.

Answer and correction;

Thank you for careful review. We agree with the reviewer. Present study included patients who referred to NK function test for suspected adult secondary HLH. HLH was suspected by physicians in cases of an unexplained fever and cytopenia. The confirmation of a suspected HLH case was performed by the diagnostic criteria in HLH-2004 guideline. According to the reviewer’s comment, we clarified that as “HLH was suspected by physicians in cases of an unexplained fever and cytopenia. The confirmation of a suspected HLH case was performed by the diagnostic criteria in HLH-2004 guideline. All patients presented with fever or cytopenia. We finally diagnosed HLH among those patients who met 5 or more HLH 2004 criteria (1, 4). Parameters included in the diagnostic guidelines for HLH were fever, splenomegaly, cytopenia affecting at least two lineages, triglycerides, fibrinogen, ferritin, and NK-cytotoxicity.” in lines 266-271. About the healthy control population, we added description in method section (lines 276-278) as “As a control for NK-cytotoxicity and NKA-IFNr assay, seventy-eight healthy donors without a history of autoimmune disorders, major infection and other inflammatory diseases were included.”

There are several contradictory statements throughout the manuscript:

Line 63: Many publications have reported alternative flowcytometric methods. However, flowcytometric-based NK cytotoxic assays have not been standardized across laboratories. They are not available outside of specialized reference centers

If they are unavailable, what value does it add to use them in this manner?

Answer and correction;

We agree with the reviewer. Although flowcytometry-based NK cytotoxicity need standardization, this assay has advantages compared to chromium release assay. We evaluated the NKA-IFNg test as an alternative test in clinical laboratories where NK cytotoxicity testing is not available. The NKA-IFNg test is easy to perform and interpret than the NK cytotoxicity, so it can be used as a supplementary diagnostic assay in patients with suspected HLH. Therefore, we described that in conclusion as “We expect that NKA-IFNr assay is a convenient and supportive screening test for HLH. Compared to NK cytotoxicity, NKA-IFNr showed similar performance for predicting HLH. It also showed high level of agreement with NK-cytotoxicity. Further validation studies are needed for NK cell function tests in large population and the prognostic value of NKA-IFNr.” In lines 259-262.

Line 79: Of a total of 119 adult patients, 50 (42.0%) patients were clinically diagnosed with HLH based on the HLH-2004 diagnostic criteria. The introduction states this as a rare disease, but 42% is not rare.

Answer and correction;

We understand reviewer’s concern. If clinicians do not have clinical suspicion of HLH and lack specific knowledge about HLH, the prevalence of HLH may be under- or overestimated, especially in adult patients. Our institute is a tertiary hospital, providing advanced treatment for patients with hematologic malignancy. In addition, the most common cause of secondary HLH was a hematologic malignancy, resulting in the high prevalence in present study. We added the description as ”HLH is a rare disease and hematologic malignancies are reported to be the main trigger of HLH (3). If clinicians do not have clinical suspicion of HLH and lack specific knowledge about HLH, the prevalence of HLH may be under- or overestimated, especially in adult patients (3, 21). In present study, all patients enrolled in this study had fever and cytopenia, therefore, the incidence of disease showed higher than expectation.” in lines 250-255.

Line: 115 There was no correlation between quantitative results from NK-cytotoxicity and NKA-IFNr (r =0.088, p > 0.05). With cutoff value of 38.5% for NK-cytotoxicity and 250 pg/mL for NKA-IFN agreement between the two assays. Is there correlation or not?

Answer and correction;

There was no correlation between quantitative data. However, qualitative results (positive or negative) with the cutoff values showed agreement. According to the reviewer’s advice, we clarified that as “When we evaluate the qualitative results with the cutoff value of 38.5% for NK-cytotoxicity and 250 pg/mL for NKA-IFNr, agreement between the two assays was 88.0% in HLH group and 58.0% in non-HLH group” in lines 138-140.

Lastly, there are several improvements needed to make the manuscript more readable.

1) Table 1: Identify what the values are that are presented, see Table.

Answer and correction;

We added value (the median with 95% CI) for each result in table 1.

2) Table 1: Causes seems to be the wrong wording since you can't have a cause for a non-HLH "disease"

According to the reviewer`s comment, we changed “causes” into “etiologic causes” in overall manuscript and table.

3) Although senestitivity is presented in Table 4, no where in the manuscript is sensitivity discussed. The authors focus solely on the specificity which when using these two methods plus ferritin (where did that come from) increases to 95% specificity but reduces sensitivity to 28%. That trade-off must be discussed and justified.

Thank you for careful review. According to the reviewer’s comment, we added description in discussion as “With cutoff value of 38.5% for NK-cytotoxicity and 250 pg/mL for NKA-IFN, the two tests showed sensitivities of 96.0% and 92.0% for diagnosing HLH, respectively. However, specificities of NK-cytotoxicity and NKA-IFNr were low (36.2% and 17.4%, respectively). To improve the specificity of NK cell function test, we combined NK cell function results and ferritin levels. Combined NKA-IFNr and ferritin levels (> 10,000 µg/L) improved specificity to 94.2%. In addition, combined NK-cytotoxicity, NKA-IFNr, and ferritin levels (> 10,000 µg/L) showed specificity of 95.7%. In practice, there is a trade-off between sensitivity and specificity. Still, no single laboratory parameter is sufficient to diagnosis HLH and superiority of current diagnostic criteria is still investigated. Thus, further new findings could be incorporated in diagnostic and prognostic marker of adult HLH.” In lines 188-197.

Round 2

Reviewer 1 Report

I feel the authors have made little attempt to address my suggestions other than a grammatical change nor do I fully accept some of the responses they have given. For example, it is universally accepted that CD56dim NK cells are more cytotoxic than CD56 bright NK cells and so it would seem reasonable (at least to me) to see if the cytotoxicity assays from donors are influenced by those donors with more (or less) CD56dim NK cells when using whole blood for their cytotoxicity assays. The same is true for CD56 bright NK cells and cytokine secretion. It is possible to try and normalise for these populations and so I do wonder why the authors don't at least attempt the analyses I suggested to see if their data follow this accepted model? This is somewhat disappointing since the authors have access to the data. In light of this, I feel my review must remain similar to the previous round.

Author Response

Thank you for this suggestion. According to the reviewer`s comment, we normalized NK cytotoxicity and NKA-IFNγ results based on NK cell (%), CD56bright NK cell (%) and CD56dim NK cell (%). Data is shown in supplementary figure S4. In addition we added description in result section as “We normalized NK cytotoxicity and NKA-IFNγ based on the percentage of NK cells, CD56bright NK cells and CD56dim NK cells (Supplementary figure S4). After normalization to CD56dim NK cells, NK cytotoxicity was observed to be significantly decreased in HLH patients compared with non-HLH patients [21.0 (13.3-24.6) vs. 31.6 (27.0-38.9), p < 0.001]. However, no significant differences were observed in NKA-IFNγ results.” in line 107-111 and in discussion as “When we normalize NK cytotoxicity data to CD56dim NK cells (%), HLH patients showed significantly decreased NK cytotoxicity levels. However, normalized NKA-IFNγ results did not show significant differences between HLH and non-HLH patients. Further validation studies are needed for NKA-IFNr tests in association with NK subpopulation.” In line 267-271.

Reviewer 2 Report

Dear Authors,

I acknowledge that the alterations in the manuscript result in an improved version, where claims are  supported by the data. A few issues remain, please provide at least a representative dot plot for point 1:

1-Authors seem to have misinterpreted my previous comment, more than what is now supplied in supplementary figures (although these do help), I was aiming to see a real dot plot explaining the NK sup-populations. Not necessarily to MFI values but to an example of quadrants/gates used to define NK population such as "CD56dim", even if it is a well known phenotype. 

2-Although authors honestly accepted my criticism and clearly state in the text limitations of the test used, it remains that one major advance proposed (a flow cytometry test for IFNg that could be used for standardisation) is unable to measure to its own announced sensitivities. Nevertheless the study does point to an added value of the use of any good similar test for these purposes. 

Minor points:

Line 42: "finding" is not needed.

Line 51: "leading" instead of lead.

Best regards

Author Response

Dear Authors,

I acknowledge that the alterations in the manuscript result in an improved version, where claims are supported by the data. A few issues remain, please provide at least a representative dot plot for point 1:

1-Authors seem to have misinterpreted my previous comment, more than what is now supplied in supplementary figures (although these do help), I was aiming to see a real dot plot explaining the NK sup-populations. Not necessarily to MFI values but to an example of quadrants/gates used to define NK population such as "CD56dim", even if it is a well known phenotype.

According to the reviewer`s comment, we added gating strategy for NK subset analysis in figure 2.

2-Although authors honestly accepted my criticism and clearly state in the text limitations of the test used, it remains that one major advance proposed (a flow cytometry test for IFNg that could be used for standardisation) is unable to measure to its own announced sensitivities. Nevertheless the study does point to an added value of the use of any good similar test for these purposes.

Thanks for your comment. This improved the quality of the paper. According to the comment, we added the limitation of our study in line 286-287 “It might be worthwhile to conduct the intracellular cytokine staining to compare the predictability of NK function assay (38,39)”.

Minor points:

Line 42: "finding" is not needed.

According to the reviewer`s comment, we corrected in line 42 as “Not all HLH patients exhibit hemophagocytosis.”.

Line 51: "leading" instead of lead.

According to the reviewers’ (reviewer 2 and 3) comments, we corrected line 50-52 as “The pathophysiological mechanism of HLH is mainly due to defective function of NK cells and cytotoxic T lymphocytes (CTLs). It results in uncontrolled activation of lymphocytes and macrophages which induce excessive production of cytokines.”

Reviewer 3 Report

Line 49 should include the phrase: of which 5 or more must be met for a diagnosis of HLH.

Lines 50-52 is a run-on sentence

Line 65: Do the authors propose that this be used outside of a specialized reference center or is its use restricted to those centers.

Line 79 should read: Of a total of 119 referred adult patients with unexplained fever and cytopenia,

Supplemental Figure 2 is difficult to interpret with the data points grouped this way. Please change so that HLH and non-HLH are next to each other for each cytokine as in Supplementary Figure 1.

Line 137 should read: There was no correlation between overall quantitative results...

Author Response

Line 49 should include the phrase: of which 5 or more must be met for a diagnosis of HLH.

According to the reviewer`s comment, we include the phrase in line 49.

Lines 50-52 is a run-on sentence

According to the reviewer`s comment, we corrected the sentence as “The pathophysiological mechanism of HLH is mainly due to defective function of NK cells and cytotoxic T lymphocytes (CTLs). It results in uncontrolled activation of lymphocytes and macrophages which induce excessive production of cytokines.”

Line 65: Do the authors propose that this be used outside of a specialized reference center or is its use restricted to those centers.

To avoid confusion, we clarified that in line 66 as “They are generally only available in specialized reference centers”.

Line 79 should read: Of a total of 119 referred adult patients with unexplained fever and cytopenia,

According to the reviewer`s comment, we corrected in line 80.

Supplemental Figure 2 is difficult to interpret with the data points grouped this way. Please change so that HLH and non-HLH are next to each other for each cytokine as in Supplementary Figure 1.

According to the reviewer`s comment, we corrected Supplementary Figure 2.

Line 137 should read: There was no correlation between overall quantitative results...

According to the reviewer`s comment, we corrected in line 162.